# *asb5a*/*asb5b* Double Knockout Affects Zebrafish Cardiac Contractile Function

**DOI:** 10.3390/ijms242216364

**Published:** 2023-11-15

**Authors:** Wanwan Cai, Yuequn Wang, Ying Luo, Luoqing Gao, Jian Zhang, Zhigang Jiang, Xiongwei Fan, Fang Li, Yulian Xie, Xiushan Wu, Yongqing Li, Wuzhou Yuan

**Affiliations:** The Laboratary of Heart Development Research, College of Life Science, Hunan Normal University, Changsha 410081, China; 201202141183@hunnu.edu.cn (W.C.); yuequnwang@hunnu.edu.cn (Y.W.); 202120142668@hunnu.edu.cn (Y.L.); 202220142836@hunnu.edu.cn (L.G.); zhangj@hunnu.edu.cn (J.Z.); 201201140149@hunnu.edu.cn (Z.J.); 16119@hunnu.edu.cn (X.F.); li-evans@hotmail.com (F.L.); xylian@hunnu.edu.cn (Y.X.); xiushanwu2003@aliyun.com (X.W.)

**Keywords:** *asb5a*, *asb5b*, CRISPR/Cas9, morpholino, RNA-seq, semi-automatic heartbeat analysis, hub genes

## Abstract

Ankyrin repeat and suppression-of-cytokine-signaling box (Asb) proteins, a subset of ubiquitin ligase E3, include Asb5 with six ankyrin-repeat domains. Zebrafish harbor two *asb5* gene isoforms, *asb5a* and *asb5b*. Currently, the effects of *asb5* gene inactivation on zebrafish embryonic development and heart function are unknown. Using CRISPR/Cas9, we generated *asb5a*-knockout zebrafish, revealing no abnormal phenotypes at 48 h post-fertilization (hpf). In situ hybridization showed similar *asb5a* and *asb5b* expression patterns, indicating the functional redundancy of these isoforms. Morpholino interference was used to target *asb5b* in wild-type and *asb5a*-knockout zebrafish. Knocking down *asb5b* in the wild-type had no phenotypic impact, but simultaneous *asb5b* knockdown in *asb5a*-knockout homozygotes led to severe pericardial cavity enlargement and atrial dilation. RNA-seq and cluster analyses identified significantly enriched cardiac muscle contraction genes in the double-knockout at 48 hpf. Moreover, semi-automatic heartbeat analysis demonstrated significant changes in various heart function indicators. STRING database/Cytoscape analyses confirmed that 11 cardiac-contraction-related hub genes exhibited disrupted expression, with three modules containing these genes potentially regulating cardiac contractile function through calcium ion channels. This study reveals functional redundancy in *asb5a* and *asb5b*, with simultaneous knockout significantly impacting zebrafish early heart development and contraction, providing key insights into *asb5*’s mechanism.

## 1. Introduction

Ankyrin repeat and suppressor-of-cytokine-signaling (SOCS) box (Asb) proteins are part of the ubiquitin ligase E3 protein family. The Asb protein subfamily, featuring six ankyrin repeat domains (Asb5, Asb9, Asb11, and Asb13), draws interest owing to unusually strong evolutionary conservation [1]. The *asb5* gene is highly homologous in higher vertebrates, such as humans, mice, and rabbits; it is a single-copy gene and is not typed [2]. However, in zebrafish, the *asb5* gene is divided into two subtypes: *asb5a* and *asb5b*. In 2003, Asb5 was identified as a novel protein linked to arteriogenesis [2]. Additionally, omics analysis revealed the close association of the *Asb* gene family, including *asb5,* with skeletal muscle development [3]. Furthermore, porcine skeletal muscle gene expression analysis indicated upregulated expression of the *asb5* gene, which is associated with muscle cell proliferation and differentiation [4]. Knockdown of *asb11* in zebrafish has prompted the formation of an enlarged pericardial cavity. All of these preliminary studies provide some clues that *asb5* may play a role in heart development. However, limited studies exist on the *asb5* gene’s role in heart development and heart function, with a 2005 study proposing its relationship with cardiovascular formation due to it as a target gene of serum response factor [5].

Zebrafish is considered important as a crucial model organism currently used to study cardiovascular development. It shares similarities of cardiovascular and hematopoietic processes with humans. During the early embryonic developmental stages of zebrafish, its body color is transparent, and the shape and function of the heart can be visually observed in living embryos. However, so far, there have been, despite this advantage, to the best of our knowledge, no reports of any existing studies on *asb5* function in zebrafish. By consulting the database, we found that *asb5* is divided into two subtypes, *asb5a* and *asb5b*, in zebrafish. In this article, using CRISPR/Cas9 genome editing technology [6,7,8], we successfully established three *asb5a* knockout zebrafish lines. The results showed that *asb5a* knockout homozygotes had, and subsequently found, no obvious abnormal phenotypes during early embryonic development. Subsequently, whole-body embryonic in situ hybridization [9] was performed, and the results showed highly similar *asb5a* and *asb5b* expression patterns in early zebrafish development. We speculated that these two genes may have potential functional redundancy, resulting in an absence of malformation when knocking out *asb5a* alone. Using morpholino interference technology [10], we successfully obtained the *asb5a*^−/−^:*asb5b*-ATGmo double-knockout zebrafish strain. RNA sequencing [11,12] was used to analyze the transcriptome differences between the *asb5a*^−/−^:*asb5b*-ATGmo double-knockout group and the wild-type (WT) group, as well as the *asb5a*^−/−^ single-knockout group, respectively. We found that the double-knockout group was significantly enriched in functional items related to cardiac muscle contraction. Semi-automatic heartbeat analysis [13] highlighted significant changes in cardiac function indicators, including notable decreases in heart rate and cardiac fractional shortening, in the *asb5a*^−/−^*:asb5b*-ATGmo double-knockout group, which were verified via quantitative polymerase chain reaction (qPCR) results. 

Using the STRING database, Cytoscape, and the Kyoto Encyclopedia of Genes and Genomes (KEGG) pathway analyses, all genes under the cardiac muscle contraction entry in the *asb5a*^−/−^:*asb5b*-ATGmo double-knockout group were compared with genes in the other two groups, leading to the production of an accurate protein–protein interaction (PPI) regulatory network. The top 20 hub genes from the two groups were screened, and 11 hub genes were identified which were closely linked to cardiac contraction in the *asb5a*^−/−^:*asb5b*-ATGmo double-knockout group: *acta1a*, *actc1a*, *actc1c*, *cacng2a*, *cacna2d1a*, *cacnb2a*, *cox7c*, *cox8b*, *uqcrc1*, *tnnc1a*, and *myl4*. Based on the combination of PPI network analysis and literature data, we hypothesized that these genes form three modules regulating early heart development in zebrafish, potentially influencing cardiac contractile function by regulating calcium ion channels. Our results suggest functional redundancy between *asb5a* and *asb5b*, as their simultaneous knockout significantly affected early zebrafish heart development and contraction. Hence, our findings provide a crucial foundation for investigating the mechanism of the *asb5* gene in zebrafish early development.

## 2. Results

### 2.1. Construction of asb5a-Knockout Zebrafish Line

To investigate the role of the *asb5a* gene in early zebrafish development, we developed an *asb5a*-knockout zebrafish strain. A pair of small guide RNAs (sgRNAs), namely Target 1 and Target 2 (Figure 1A), were designed for exon 1 of *asb5a*, resulting in threezebrafish lines (Lines 1–3) with frameshift mutant alleles. Sanger sequencing revealed specific mutations: Line 1, 8 bp insertion and 174 bp deletion; Line 2, 172 bp deletion; and Line 3, 175 bp deletion (Figure 1B,C). F2 homozygotes, derived from self-crossed heterozygotes, exhibited significantly downregulated *asb5a* mRNA expression (Appendix A) but no apparent phenotypic differences compared with the WT. This led us to explore the potential functional redundancy between *asb5a* and *asb5b*. We hypothesized that *asb5b* gene expression would compensate for the deficiency caused by complete *asb5a* gene knockout.

### 2.2. Whole-Embryo In Situ Hybridization Reveals the Similar Early Expression Patterns of asb5a and asb5b

As no observable embryonic malformations occurred in *asb5a* homozygous knockout lines, we hypothesized that this was due to functional compensation by *asb5b* in vivo. Whole-embryo in situ hybridization(WISH) experiments revealed the consistent expression patterns of *asb5a* (Figure 2A–E) and *asb5b* (Figure 3A–E) from early development to 4 days post-fertilization (dpf). Both genes exhibited similar expression patterns during early stages, with ubiquitous expression starting at 48 hours post-fertilization (hpf), and expression particularly concentrated in the eyes and head areas. Expression gradually decreased as the embryo developed, up to 4 dpf.

### 2.3. Construction of asb5a^−/−^:asb5b-ATGmo Double-Knockout Zebrafish Strain

To confirm that *asb5a* knockout alone did not induce malformation due to *asb5b* functional compensation in vivo, we performed specific knockdown of *asb5b* in *asb5a*^−/−^ embryos and WT embryos, respectively. Morpholino oligos specifically targeting *asb5b* were produced in GeneTools to block transcription (*asb5b*-ATGmo: 5′-TAACCACAATGTGTGCCTACTTACT-3′; Appendix A). Injecting WT one-cell stage embryos with 0.25Mm morpholino oligos effectively reduced *asb5b* mRNA expression (Appendix A–D). An *asb5b*-ATGmo concentration gradient was used to determine the optimal injection concentration (Figure 4B–E). Injecting *asb5b*-ATGmo at the one-cell stage in the WT led to no obvious phenotype malformation or embryonic lethality at 48 hpf, regardless of low or high morpholino concentrations (Appendix A; Appendix A). However, injecting *asb5b*-ATGmo at the one-cell stage in *asb5a*^−/−^ homozygous embryos(hereinafter, *asb5a*^−/−^:*asb5b*-ATGmo double-knockout zebrafish) resulted in pericardial cavity enlargement and atrial dilation at 24 hpf, progressing to 48 hpf (Figure 4G–G″). Malformation severity increased with higher *asb5b*-ATGmo concentrations, with 0.75 mM considered the optimal concentration for subsequent experiments.

### 2.4. Transcriptome Sequencing Data Analysis

As the *asb5a^−/−^*:*asb5b*-ATGmo double-knockout zebrafish displayed severe cardiac malformations by 48 hpf, high-throughput transcriptome sequencing was used to compare the WT, *asb5a^−/−^* homozygous, and *asb5a^−/−^*:*asb5b*-ATGmo double-knockout groups at 48 hpf. Relative to the WT, the *asb5a^−/−^* group exhibited 879 upregulated and 1164 downregulated differentially expressed genes (DEGs). The *asb5a^−/−^:asb5b*-ATGmo double-knockout group had 5766 upregulated and 4469 downregulated DEGs. Compared with the *asb5a^−/−^* group, the double-knockout group had 6354 upregulated and 4233 downregulated DEGs (Figure 5A–C). All DEGs were shown in heatmaps in Figure 5D–F. Gene Ontology(GO) and KEGG pathway enrichment analyses revealed significant biological functions and pathways affected by the *asb5a/asb5b* deletion. Figure 6A–C showed the 20 most important GO entries, which included biological regulation (GO:0065007; biological process), catalytic activity (GO:0003824; molecular function), and cell part (GO:0044464, cellular component). In addition, the top 20 KEGG pathways included the MAPK signaling pathway, cardiac muscle contraction, and human papillomavirus infection (Figure 7A,B). Notably, *asb5a* knockout alone did not significantly impact any signaling pathways compared with the WT, further highlighting the potential limited effect of *asb5a* gene deletion in isolation.

### 2.5. Early Cardiac Contraction Abnormalities Induced by asb5a^−/−^:asb5b-ATGmo Double-Knockout in Zebrafish

A detailed RNA-seq analysis revealed an abundance of cardiac muscle contraction pathways in KEGG pathway enrichment analysis regarding the *asb5a^−/−^:asb5b*-ATGmo double-knockout group. This enrichment persisted compared with both WT and *asb5a^−/−^* groups (illustrated in the purple box in Figure 7A,B). Thus, we hypothesized that concurrent knockout of the *asb5a* and *asb5b* genes in zebrafish impacts early cardiac contractile function. Using a high-speed electron-multiplying charge-coupled device (EM-CCD) camera, we recorded 10 s videos at 130 frames s^−1^. Subsequently, semi-automatic heartbeat analysis software (Version 3.4.0.0) was employed to analyze the recorded heartbeat movies, generating various indicators related to cardiac contraction, including heart rate, heart period (HP), diastolic interval (DI), systolic interval (SI), diastolic diameter (DD), systolic diameter (SD), and fractional shortening(FS). All data were generated in M-mode. Approximately 30 zebrafish embryos at 48 hpf in each group were randomly selected and imaged under a 20× lens. Focusing on the atrial tissue, given the previously observed atrial dilation in the *asb5a^−/−^:asb5b*-ATGmo double-knockout group in the *myl7*:EGFP strain, we photographed the atrium in M-mode to assess various aspects of this tissue. Cross-sectional images from the 10 s heartbeat movie enabled intuitive analysis of DD, SD, SI, HP, and DI for each group (Figure 8A). Compared with the WT group and the single-gene-knockout group, the *asb5a^−/−^:asb5b*-ATGmo double-knockout group exhibited a significantly lower heart rate (Figure 8B) and markedly prolonged HP, SI, and DI (Figure 8C–E). The *asb5a^−/−^* group and *asb5b*-ATGmo group, in contrast to the WT group, showed an upward trend in heart rate (Figure 8B), along with significantly lower HP and DI (Figure 8C,E), whereas SI showed no apparent change (Figure 8D). No significant differences in various indicators were observed between the *asb5a^−/−^
*group and the *asb5b*-ATGmo single-gene–knockdown group (See the Appendix A for details).

By capturing individual frames of the video (image sizes:1300 × 500), we observed a notable increase in the DD/SD of the atrium in the *asb5a^−/−^:asb5b*-ATGmo double-knockout group compared with the other groups (Figure 9A–C). This suggests an evident trend toward atrial expansion. In contrast, the single-gene-knockout group showed no significant difference in DD/SD compared with the WT group (Figure 9A–C). No significant variations in any indicators were detected between the *asb5a^−/−^* group and the *asb5b*-ATGmo single-gene-knockdown group. We calculated the relative value of cardiac fractional shortening using the following formula: FS = (DD − SD)/DD × 100%. FS represents the ejection volume at each contraction of the heart. Reduced FS indicates the inability of the heart to contract completely during systole, corresponding to systolic heart failure in the human adult heart. Statistical results revealed a significantly lower cardiac fractional shortening in the *asb5a^−/−^:asb5b*-ATGmo double-knockout group compared with the other groups (Figure 9D) (See the Appendix A for details).

### 2.6. PPI Network Construction and Hub Gene Screening

In a more in-depth examination of RNA-seq data, using the KEGG pathway enrichment analysis, it was evident that the *asb5a^−/−^:asb5b*-ATGmo double-knockout group exhibited pronounced enrichment of cardiac muscle contraction entries compared with other groups (Figure 7A,B; highlighted in purple). Under these entries, the *asb5a^−/−^:asb5b*-ATGmo double-knockout group showed enrichment in 87 genes compared with the WT group, and 91 genes compared with the *asb5a^−/−^* group. Initially, the STRING database was employed to analyze the network diagram of unified regulatory genes in these two sets of comparative experiments. Subsequently, using Cytoscape software (Version 3.9.1), the connectivity (degree) was organized from high to low, producing a precise PPI regulatory network diagram (Figure 10A,C). By sorting from high to low according to connectivity (degree), the top 20 hub genes were identified, and their expression patterns were visualized in a heatmap based on RNA-seq data (Figure 10B,D).

### 2.7. The qPCR Validation of 20 Abnormally Regulated Hub Genes

To corroborate whether the expression levels of the top 20 hub genes between the two groups aligned with the transcriptome data, qPCR experiments were conducted. The results revealed that, in comparison to the WT group, six genes (*acta1a*, *actc1a*, *actc1c*, *cacng2a*, *cacna2d1a*, and *cox7c*) in the *asb5a^−/−^:asb5b*-ATGmo double-knockout group were significantly upregulated, and *cacnb2a* was significantly downregulated. In comparison to the *asb5a^−/−^* group, five genes (*acta1a*, *cox8b*, *uqcrc1*, *tnnc1a*, and *myl4*) in the *asb5a^−/−^:asb5b*-ATGmo double-knockout group were significantly upregulated. These results were consistent with the aforementioned transcriptome findings (Figure 11A,B). Through qPCR validation, overlapping genes were identified between groups, which were all differentially expressed in the *asb5a^−/−^:asb5b*-ATGmo double-knockout group compared with the control group. Subsequently, the STRING database and Cytoscape software (Version 3.9.1) were employed to construct a novel PPI network diagram, indicating the unified regulation of these 11 DEGs (*acta1a*, *actc1a*, *actc1c*, *cacng2a*, *cacna2d1a*, *cacnb2a*, *cox7c*, *cox8b*, *uqcrc1*, *tnnc1a*,and *myl4*). The PPI network contained three modules with 11 nodes and 13 edges (Figure 11C).

## 3. Discussion

In this study, zebrafish served as the animal model, and we successfully established an *asb5a^−/−^:asb5b*-ATGmo double-knockout line. This research elucidated, for the first time, the crucial role of the *asb5* gene in early cardiac morphogenesis and function in zebrafish, suggesting potential functional redundancy between *asb5a* and *asb5b*.

Initially, we employed CRISPR/Cas9 technology for successful *asb5a* gene knockout in zebrafish. However, no apparent phenotype was observed in the homozygous mutant. We hypothesized that the lack of observable effects stemmed from the division of the *asb5* gene into two subtypes, *asb5a* and *asb5b*, with the latter compensating for defects caused by *asb5a* homozygosity. Subsequently, using morpholino interference technology, we further knocked down the *asb5b* gene in the *asb5a* homozygous mutant. This revealed significant atrial amplification malformations in embryos with simultaneous knockout of these two genes. Through RNA-seq cluster analysis, we targeted cardiac muscle contraction-related functions. Semi-automatic heartbeat analysis software (Version 3.4.0.0) revealed disordered cardiac function indicators in the double-knockout strain, impacting early zebrafish cardiac contractility. Further analysis of the genes enriched in KEGG under the cardiac muscle contraction entry, conducted through the STRING database, Cytoscape software (Version 3.9.1), and qPCR experiments, led to the identification of 11 hub genes (*acta1a*, *actc1a*, *actc1c*, *cacng2a*, *cacna2d1a*, *cacnb2a*, *cox7c*, *cox8b*, *uqcrc1*, *tnnc1a*, and *myl4*). Their expression disorders may directly or indirectly regulate early cardiac contractile function in zebrafish.

Myosin, as the primary component of thick myofilaments, mediates force generation during muscle contraction by interacting with actin filaments [14]. The N-terminus of the cardiac myosin light chain can regulate cardiac contractile function by binding to the C-terminus of actin [15]. Additionally, troponin may interact with actin, myosin, Ca^2+^-binding proteins, and tropomyosin, inhibiting actomyosin ATPase and the movement of actin filaments on myosin in vitro [16]. The troponin complex serves as a molecular switch linking changes in intracellular calcium concentration to actin and myosin association and dissociation, allowing excitation–contraction coupling in striated muscle [17]. Troponin may act as a mediator between calcium ion channels and myosin/actin, effectively regulating cardiac muscle contraction. Cardiomyocyte contraction is triggered by the entry of extracellular calcium through specific calcium channels [18]. L-type calcium channels (LTCCs) are abundant in the myocardium. Voltage-gated LTCCs are located on the cardiomyocyte surface and contribute to cardiac action potentials by mediating calcium entry into the cell and triggering calcium-induced calcium release. Calcium voltage-gated channel auxiliary subunit β 2 (CACNB2) is a membrane-associated guanylate kinase protein, and the CACNB2 LTCC auxiliary subunit transports pore-forming CACNA subunits to the membrane and regulates channel dynamics [19]. Voltage-gated calcium ion channels consist of pore-forming α(1) and auxiliary α(2)δ, β, and γ subunits [20]. CACNG2 (gamma 2) may regulate calcium channel homeostasis by interacting with CACNA and CACNB. Finally, cytochrome c oxidase (COX), the rate-limiting enzyme of mitochondrial respiration, is regulated by various mechanisms [21]. COX is the terminal enzyme of the electron transport chain and catalyzes the transfer of electrons from cytochrome c to oxygen. Cox7c, Cox7b, ATP synthase, complex III subunit X, and myosin heavy chain 7 may be involved in cardiovascular metabolic pathways [22]. Cox7c and Cox8b potentially regulate oxygen consumption and ATP production. Ubiquitin cytochrome c reductase core protein 1 (UQCRC1) is an integral component of mitochondrial complex III, playing a key role in cardiac protection and maintaining mitochondrial function [23].

Through analysis of the preliminary PPI network shown in Figure 11C and a review of existing literature, we constructed a probable molecular regulatory network affecting early cardiac contractile function in zebrafish in the *asb5a^−/−^:asb5b*-ATGmo double-knockout line (Figure 12). We speculate that it may impact early cardiac contractile function in zebrafish through three pathways: first, inactivation of *asb5a* and *asb5b* genes leads to the upregulation of myosin light chain 4 (*myl4*) [24], *actc1a*, *actc1c* [25] and *acta1a* [26]. These *myl4*-associated genes are significantly upregulated, ultimately leading to increased transcription of the cardiac troponin C regulatory gene *tnnc1a* [27], potentially influencing calcium ion channels. Second, knockout of *asb5a* and *asb5b* genes results in disorders in the mRNA levels of the calcium-channel-related genes *cacng2a* [28], *cacnb2a* [19], and *cacna2d1a* [29], impacting calcium channel homeostasis. Third, deletion of *asb5a* and *asb5b* genes leads to the upregulation of the COX family genes *cox7c* [30] and *cox8b* [31], followed by increased expression of *uqcrc1* [23], potentially affecting calcium ion channel stability and leading to cardiac contractile dysfunction.

In summary, our study has preliminarily affirmed the crucial role of the *asb5* gene in early heart development in zebrafish. Its deletion disrupts the expression of several early cardiac contraction-related genes, affecting the morphogenesis and functional integrity of the early heart, providing vital insights for further investigation into the molecular regulatory mechanisms of this gene in heart development, and expanding the range of novel potential targets for understanding early congenital heart disease.

## 4. Materials and Methods

### 4.1. Zebrafish Strain Rearing and Breeding

The AB zebrafish strain and the *myl7*:EGFP strain (*myl7* is a heart-specific gene, which is especially expressed in the heart, and *myl7*:EGFP transgenic zebrafish is used as a cardiac-specific expression marker) were maintained at a constant temperature (28.5 °C) in a water circulation system. Zebrafish breeding involved placing two females and three males into hybrid bars, with overnight separation using clear plastic partitions. Partitions were removed around 09:00 the next day to facilitate mating, egg laying, and subsequent embryo collection.

### 4.2. CRISPR/Cas9-Mediated asb5a Gene Knockout

The CRISPR/Cas9 gene editing system was employed for zebrafish *asb5a* knockout. PCR was performed using the P42250 plasmid as the sgRNA template to synthesize the sgRNA transcript. The reverse primer was sgRNA-R (a universal R-terminal primer for the target site; Appendix A), and the two forward primers containing the T7 promoter and target sequence were sgRNA-primer-F1 and sgRNA-primer-F2, respectively (Appendix A). The sgRNA was synthesized via in vitro transcription using the Riboprobe^®^ System-T7 Transcription Kit (P1440, Promega, Madison, WI, USA). The sgRNAs (final concentration: 20 ng/µL each) were co-injected with Invitrogen TrueCut Cas9 v2 (A36499, Thermo Fisher Scientific, Waltham, MA, USA; final concentration: 300 ng/µL) into one-cell-stage fertilized eggs. The sgRNAs were designed to use CRISPR/Cas9 technology to introduce frameshift mutations into the coding sequence of *asb5a* through non-homologous end joining [6,32,33], thereby causing the *asb5a* frameshift mutation to produce a premature stop codon for protein translation. Surviving embryos with chimeras were mated with WT zebrafish, and offspring with partial deletion mutant alleles of *asb5a* were sequenced. Genotype identification for the *asb5a*-knockout lines was conducted using the forward and reverse primers *asb5a*-F and *asb5a*-R, respectively (Appendix A).

### 4.3. RNA Probe Synthesis and WISH

For mRNA antisense probe preparation, RT-PCR was used to amplify a gene’s mRNA sequence using primers with T7 promoter sequences (*asb5a* and *asb5b* forward and reverse primers, as detailed in Appendix A). Digoxigenin-labeled antisense RNA probes were synthesized through in vitro transcription using the Riboprobe^®^ System-T7 Transcription Kit (P1440, Promega, n vitro transcription using the Riboprobe^®^ System-T7 Transcription Kit (P1440, Promega, Madison, WI, USA) and ROCHE DIG RNA Labeling Mix (REF 11277073910, Roche, Basel, Switzerland). Zebrafish embryos at specific stages were fixed in 4% paraformaldehyde and stored in 100% methanol. WISH was used to detect *asb5a* and *asb5b* mRNA expression, and images were captured using a fluorescence microscope (Leica M205FA, Wetzlar, Germany) with Leica Application Kit imaging software (version 3.2.0).

### 4.4. Morpholino-Mediated Specific Knockdown of the asb5b Gene

A morpholino oligo sequence targeting *asb5b* was obtained using GeneTools (https://www.gene-tools.com, Philomath, OR, USA) to inhibit transcription (*asb5b*-ATGmo: 5′-TAACCACAATGTGTGCCTACTTACT-3′). Injection of a low *asb5b*-ATGmo concentration (0.25 mM) was performed at the one-cell stage in WT embryos. Four tubes of embryos from each group (WT and experimental groups) were collected (10 embryos per tube) as biological replicates, and RNA was extracted and reverse-transcribed into a cDNA library. Grayscale analysis performed via RT-PCR confirmed effective knockdown of *asb5b* mRNA levels at low concentrations; qPCR yielded the same results (Appendix A).

### 4.5. RNA-Seq-Based Transcriptome Analysis

Zebrafish embryos at 48 hpf were collected in batches of 50 embryos per tube, constituting a biological replicate; three tubes were randomly collected and stored at −80 °C. Each RNA-seq dataset was derived from a biological replicate of 50 embryos. Subsequent RNA-seq was conducted at Majorbio (Shanghai, China), with data analysis performed on the Majorbio Cloud platform. DEGs were analyzed using DESeq with screening conditions set at a significance threshold of |log2FC| > 2.0 and *p* < 0.05. RNA-seq results for the WT group, *asb5a^−/−^* group (Group A in Appendix A) and *asb5a^−/−^: asb5b*-ATGmo double-knockout group(Group B in Appendix A) are shown in Appendix A.

The volcanic map of the DEGs was drawn using three software packages, DESeq2 (version 1.57.1) [34], DEGseq (version 1.56.0) [35], and edgeR (version 4.0.1) [36]. Genes/transcripts with similar expression patterns may have similar functions or participate in the same metabolic process or cellular pathway. Based on the expression level of genes/transcripts in different samples, the distances between genes/transcripts or samples were calculated, and then iterative methods were used to cluster genes or samples. GO enrichment analysis was performed using Goatools software (https://github.com/tanghaibao/GOatools, version 0.6.9, accessed on 25 July 2023) [12], and when corrected *p* values (FDR) < 0.05, they were considered to be significantly enriched, and sufficient for GO function. KEGG PATHWAY enrichment analysis was performed using R language. The principle of calculation was similar to that of enrichment analysis of GO function, with p < 0.05 as the cut-off criterion.

### 4.6. Imaging Technique and Image Analysis

Zebrafish embryos from the *myl7*:EGFP strain were imaged using a fluorescence microscope (Leica M205FA, Wetzlar, Germany). In situ hybridization images were captured with Leica Application Kit imaging software (version 3.2.0). Live embryos, incubated in Petri dishes with fresh water at 28.5 °C, were immobilized using 6% methylcellulose for imaging. An Axiocam (Zeiss Company, Jena, Germany) was used to photograph and analyze larval morphology at 24 and 48 hpf. The pericardial cavities and cardiac phenotypes of juvenile fish at these two stages were analyzed as previously described. Cardiac atria labeled with *myl7*:EGFP were visible against a green fluorescence background, revealing a propensity for severe atrium dilation in the double-knockout line. Images were processed using Zeiss Axio Vision 3.0.6 software and Adobe Photoshop. A high-speed EM-CCD camera captured zebrafish heart activity at 48 hpf for 10 s at 130 frames s^−1^ under a 20× microscope. Semi-automatic heartbeat analysis software was then used to analyze the recorded heartbeat movie, generating indicators related to cardiac contraction: heart rate, HP, DI, SI, DD, SD, and fractional shortening (FS). All data were also reflected in the generated M-mode (See the supplementary material Excel Appendix A for details).

### 4.7. PPI Network Construction and Module Selection

PPI networks were constructed using STRING version 11.0b (https://string-db.org/; accessed on 2 June 2022) and Cytoscape (version 3.9.1). DEGs were sorted based on degree values, and the top 20 DEGs were designated as hub genes.

### 4.8. The qPCR detection

To detect gene expression, qPCR was employed. Each group comprised three biological replicates, with each replicate consisting of ten zebrafish embryos at 48 hpf and four technical replicates. Total RNA extraction involved the use of the Novozyme Universal RNA Extraction Kit (RC112-01, Vazyme, Düsseldorf, Germany), and cDNA was synthesized using the PrimeScript RT Kit (RR036A-1, Takara, Japan). Target gene expression was normalized using the internal control *actb1* (also named *β-actin*, an internal reference gene, which has a relatively constant expression in various tissues and cells, and is often used as a reference when detecting gene expression level changes). The QuantStudio 5 Real-Time PCR system (Thermo Fisher Scientific, Waltham, MA, USA) and QuantStudio design and analysis software (version 1.5.2) were employed for qPCR. Primers for target genes, including those for *acta1a, actc1a, actc1c, cacng2a, cacna2d1a, cacnb2a, cox7c, cox8b, uqcrc1, tnnc1a,* and *myl4*, are detailed in Appendix A. Relative gene expression values were calculated using the ΔΔCT method.

### 4.9. Statistical Analysis

The statistical significance of qPCR and heartbeat test data was assessed using an unpaired *t*-test with the significance level set at *p* < 0.05.

## Figures and Tables

**Figure 1 ijms-24-16364-f001:**
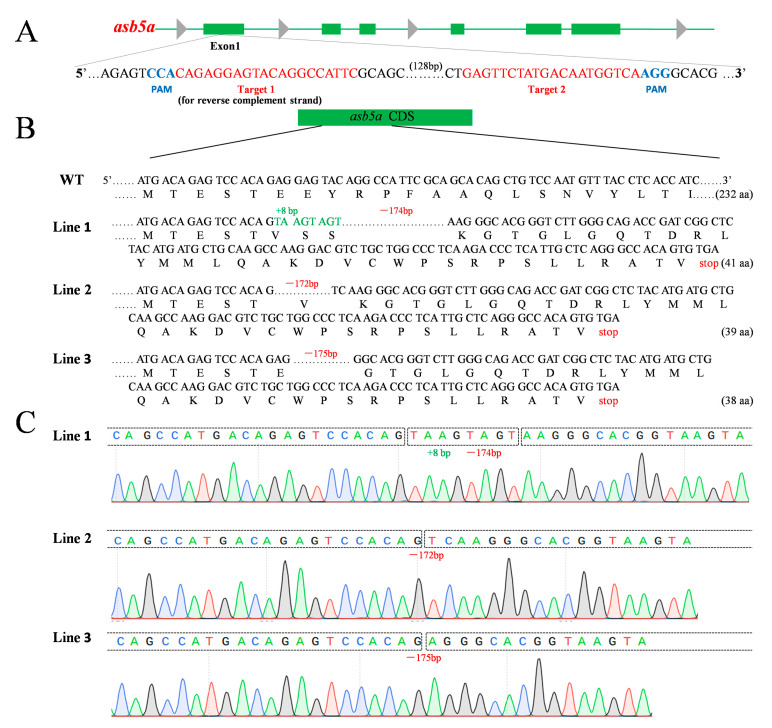
Schematic diagrams of *asb5a* gene knockout in zebrafish. (**A**) Schematic diagram of sgRNA targeting for *asb5a* gene knockout. Green horizontal line represents the genomic DNA of *asb5a*, green rectangle represents the exons of *asb5a*, red font represents the target site sequence, and blue font represents the protospacer adjacent motif (PAM). (**B**) Schematic diagram of the three heritable mutant alleles of *asb5a* produced via gene knockout and their encoded protein sequences. (**C**) Sequencing peak diagram of three *asb5a* mutant alleles, in which bases A, G, C, and T are represented by green, black, blue, and red curves, respectively.

**Figure 2 ijms-24-16364-f002:**
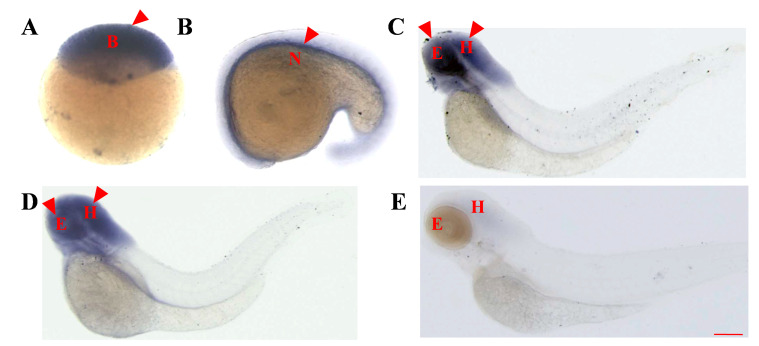
Temporal and spatial expression patterns of the *asb5a* gene in zebrafish embryos determined through whole-embryo in situ hybridization. (**A**) High blastocyst stage (3.3 hpf), with ubiquitous expression. (**B**) At the 21-somite stage (19.5 hpf), *asb5a* is expressed in the notochord region. (**C**) At 48 hpf, high expression of *asb5a* is concentrated in the eyes and head area, with low expression in the tail. (**D**) At 3 dpf, *asb5a* is highly expressed in the head and the eyes. (**E**) At 4 dpf, *asb5a* expression is downregulated in the eye and head regions. hpf: hours postfertilization; dpf: days postfertilization; B: blastocyst; N: notochord; E: eyes; H: head. Scale bar: 1 mm.

**Figure 3 ijms-24-16364-f003:**
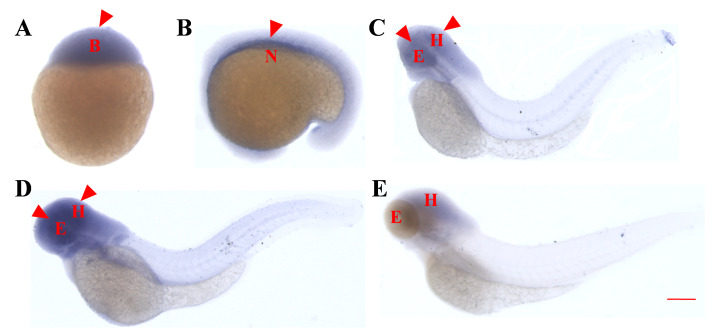
Temporal and spatial expression pattern of the *asb5b* gene in zebrafish embryos determined through whole-embryo in situ hybridization. (**A**) High blastocyst stage (3.3 hpf), with ubiquitous expression. (**B**) At the 21-somite stage (19.5 hpf), *asb5b* is expressed in the notochord region. (**C**) At 48 hpf, high expression of *asb5b* is concentrated in the eyes and head area, with low expression in the tail. (**D**) At 3 dpf, *asb5b* is highly expressed in the head and the eyes. (**E**) At 4 dpf, *asb5b* expression is downregulated in the eyes and head regions. B: blastocyst; N: notochord; E: eyes; H: head. Scale bar: 1 mm.

**Figure 4 ijms-24-16364-f004:**
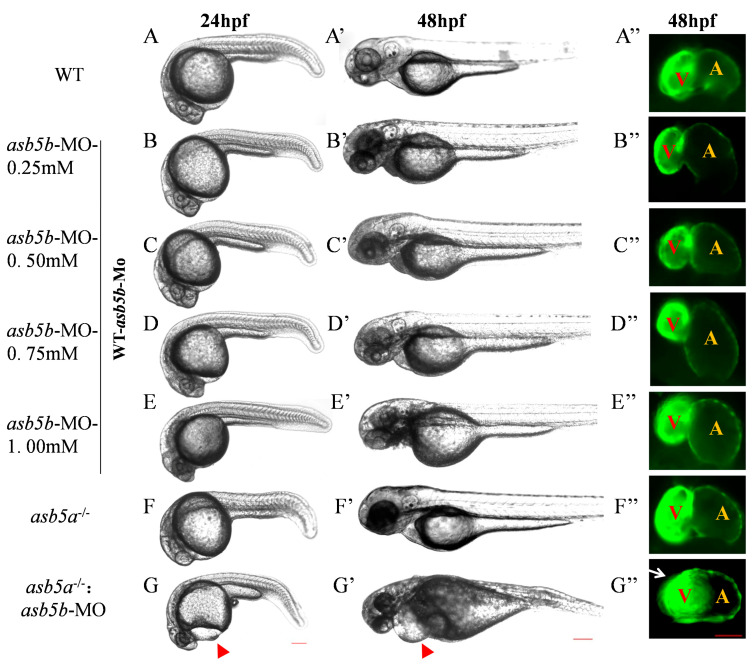
Effects of *asb5b*-ATGmo knockdown on zebrafish heart development. (**A**–**G**) Pericardial cavities of different groups when embryos had developed to 24 hpf. Scale bar: 200 µm. (**A′**–**G′**) Pericardial cavities of different groups when embryos had developed to 48 hpf. Scale bar: 200 µm. (**A″**–**G″**) For zebrafish on the *myl7*:EGFP background (*myl7*:EGFP transgenic zebrafish indicates EGFP is specially expressed in the heart, see Section 4.1), an atrium (A) and ventricle (V) from each group is shown at the point when the embryos had developed to 48 hpf. Red triangular arrows indicate the enlarged area around the cardiac cavity, and a white arrow indicates the atrial expansion area. Scale bar: 50 µm.

**Figure 5 ijms-24-16364-f005:**
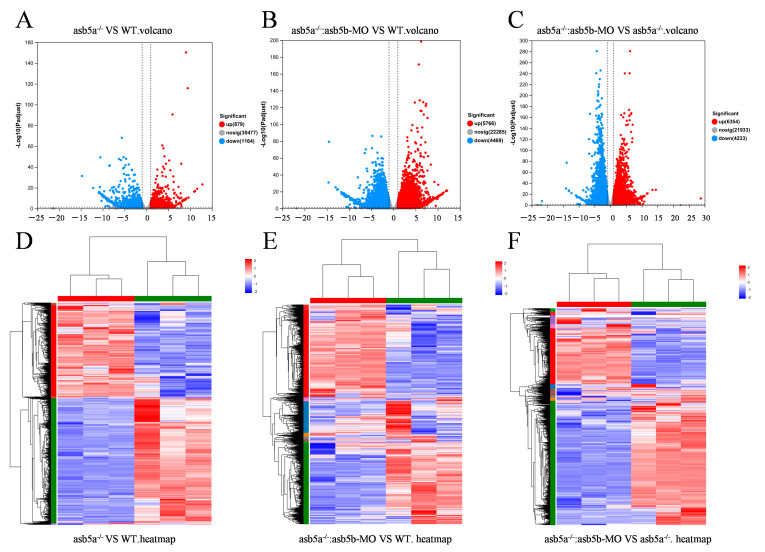
Effects of *asb5a^−/−^*:*asb5b*-ATGmo double-knockout on the transcriptome of zebrafish at 48 hpf. (**A**–**C**) Volcano plot of differentially expressed genes (DEGs). Blue dots represent downregulated genes, red dots represent upregulated genes, and gray dots represent genes with no significant expression differences under different conditions (significance threshold: |log2FC| > 2.0 and *p* < 0.05). (**D**–**F**) Heatmap of DEGs, where rows and columns represent genes and samples, respectively. Red and blue indicate high and low expression levels, respectively. Darker colors denote more pronounced significant differences.

**Figure 6 ijms-24-16364-f006:**
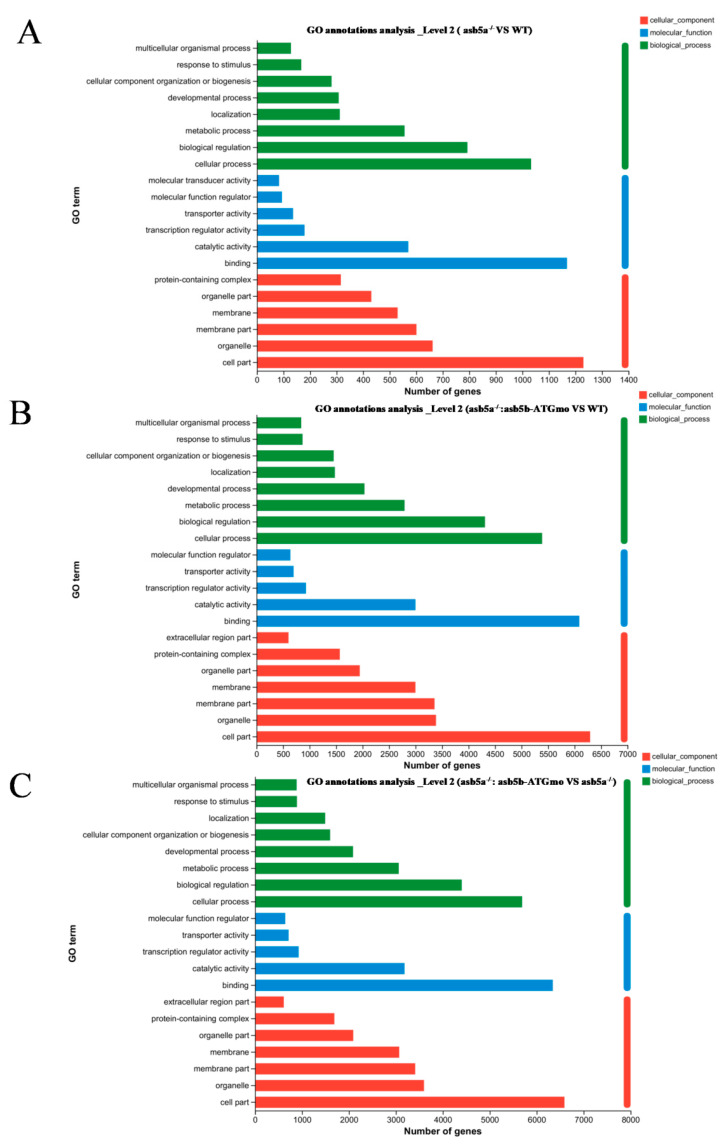
Effects of *asb5a^−/−^*:*asb5b*-ATGmo double-knockout on Gene Ontology (GO) enrichment analysis of zebrafish at 48 hpf. GO enrichment analysis of DEGs includes molecular functions (MFs), biological processes (BPs), and cellular components (CCs). GO enrichment analysis results are shown for (**A**) the *asb5a^−/−^* group compared with the wild-type (WT) group, (**B**) the *asb5a^−/−^*:*asb5b*-ATGmo double-knockout group compared with the WT group, and (**C**) the *asb5a^−/−^*:*asb5b*-ATGmo double-knockout group compared with *asb5a^−/−^* group.

**Figure 7 ijms-24-16364-f007:**
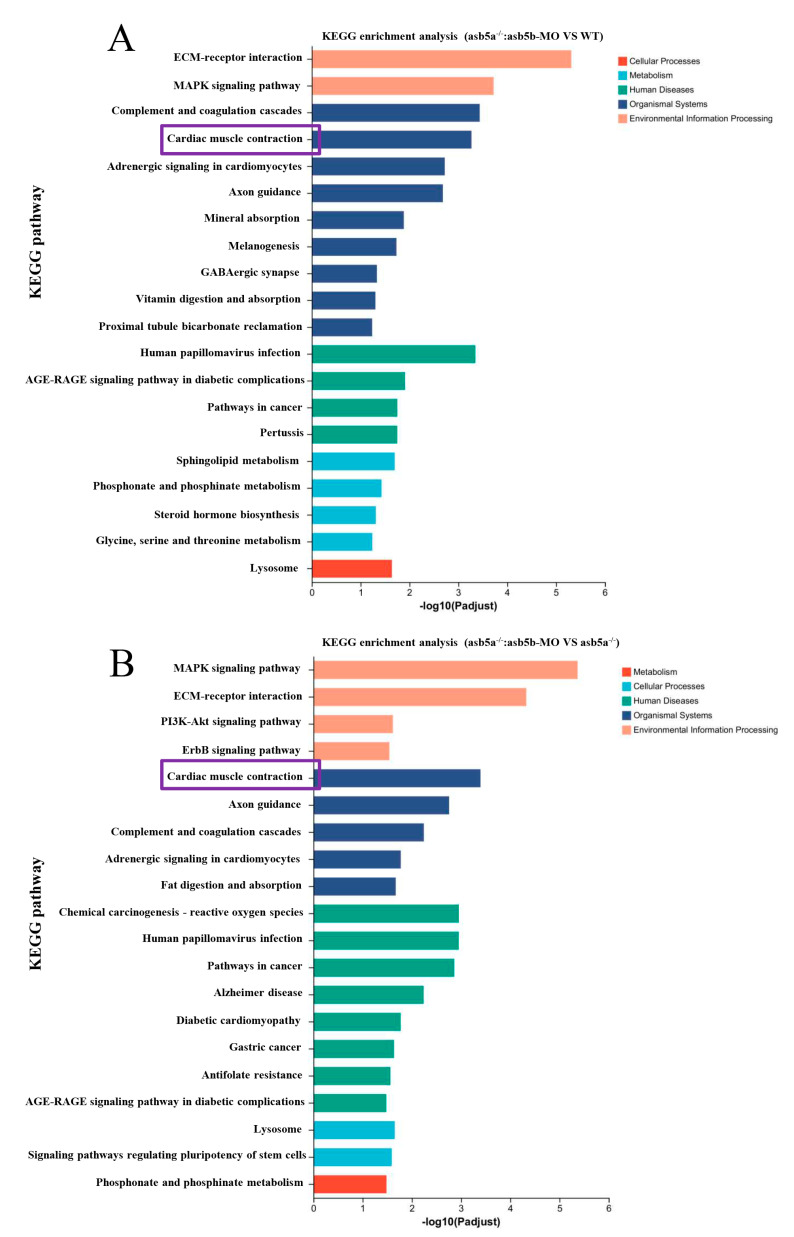
Effect of *asb5a^−/−^*:*asb5b*-ATGm double-knockout on Kyoto Encyclopedia of Genes and Genomes (KEGG) pathway enrichment analysis of zebrafish at 48 hpf. KEGG pathway enrichment analysis of DEGs includes only the top 20 functional entries. (**A**) KEGG pathway enrichment analysis results for the *asb5a^−/−^*:*asb5b*-ATGmo double-knockout group compared with the WT group. (**B**) KEGG pathway enrichment analysis in the *asb5a^−/−^*:*asb5b*-ATGmo double-knockout group compared with the *asb5a^−/−^* group. Purple box represents KEGG pathway enriched in cardiac muscle contraction function entries.

**Figure 8 ijms-24-16364-f008:**
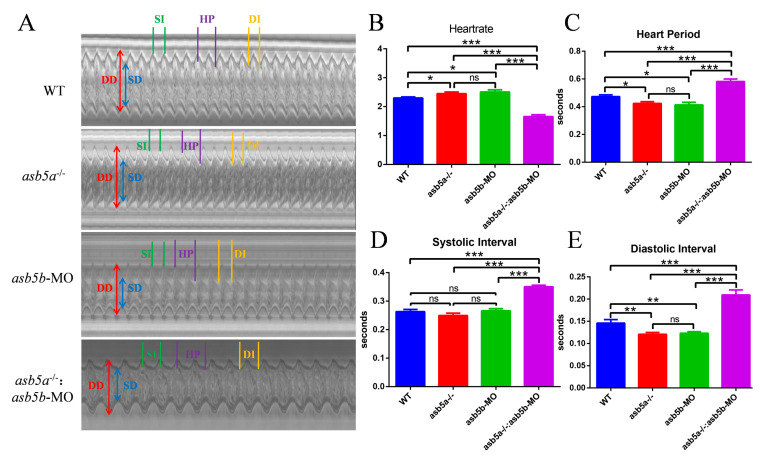
M-mode revealed cardiac physiological functions after *asb5a/asb5b* inactivation. The 10 s M-modes clearly displayed various physiological and functional indicators. (**A**) Compared with the WT group, the *asb5a^−/−^:asb5b*-ATGmo double-knockout group showed prolonged HP, SI, and DI. (**B**) Heart rate statistical results. Compared with the WT group, heart rate in the single-gene-knockout group was significantly increased, whereas heart rate in the double-gene-knockout group was significantly decreased. (**C**) Statistical results of HP. Compared with the WT group, HP in the single-gene-knockout group was significantly shortened, whereas HP in the double-gene-knockout group was significantly increased. (**D**) Statistical results of SI. Compared with the WT group, there was no significant change in SI in the single-gene-knockout group, whereas SI in the double-gene-knockout group was significantly prolonged. (**E**) Statistical results of DI. Compared with the WT group, DI in the single-gene-knockout group was significantly shortened, whereas DI in the double-gene-knockout group was significantly prolonged. HP: heart period; SI: systolic interval; DI: diastolic interval; DD: diastolic diameter; SD: systolic diameter. Plotted data represent means ± standard error of the mean (SEM; n = 30). ^ns^ *p* > 0.05; * *p* < 0.05; ** *p* < 0.01; *** *p* < 0.001.

**Figure 9 ijms-24-16364-f009:**
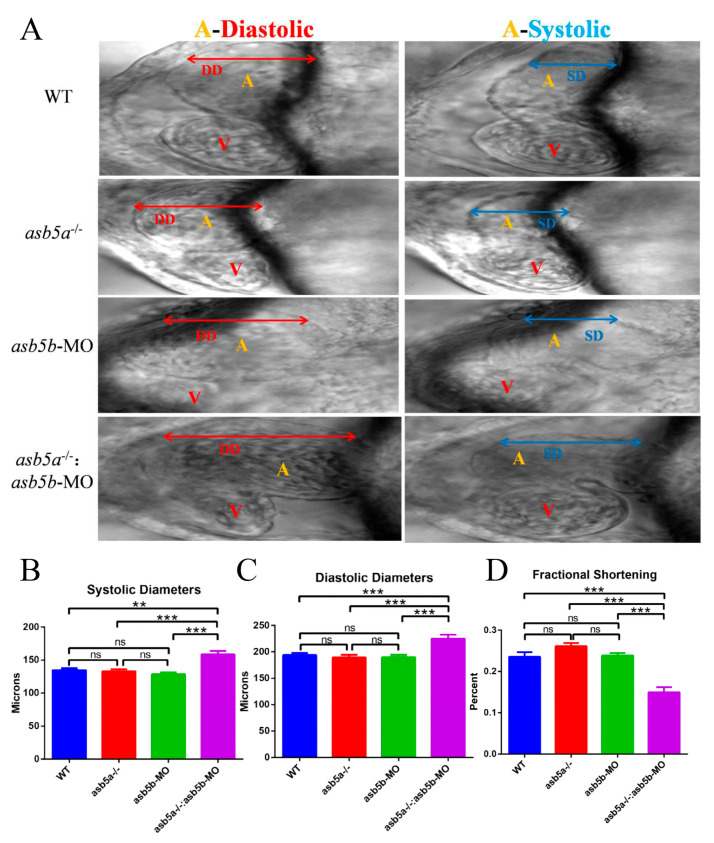
Atrial morphological changes after *asb5a/asb5b* inactivation. The 10 s M-mode screenshot clearly displayed the diastolic and systolic shapes of the atrium (A) and ventricle (V). (**A**) Morphological results. Compared with the other groups, the atrium tissue of the *asb5a^−/−^:asb5b*-ATGmo double-knockout group exhibited significantly longer DD and SD. (**B**) Statistical results of cardiac SD. Compared with the other groups, the SD of the *asb5a^−/−^:asb5b*-ATGmo double-knockout group was significantly increased. (**C**) Statistical results of DD. Compared with the other groups, the DD of the *asb5a^−/−^:asb5b*-ATGmo double-knockout group was significantly increased. (**D**) Statistical results of cardiac fractional shortening. Compared with the other groups, the cardiac fractional shortening in the *asb5a^−/−^:asb5b*-ATGmo double-knockout group was significantly reduced. DD: diastolic diameter; SD: systolic diameter. Plotted values represent means ± SEM (n = 30). ^ns^
*p* > 0.05; ** *p* < 0.01; *** *p* < 0.001 (unpaired *t*-test).

**Figure 10 ijms-24-16364-f010:**
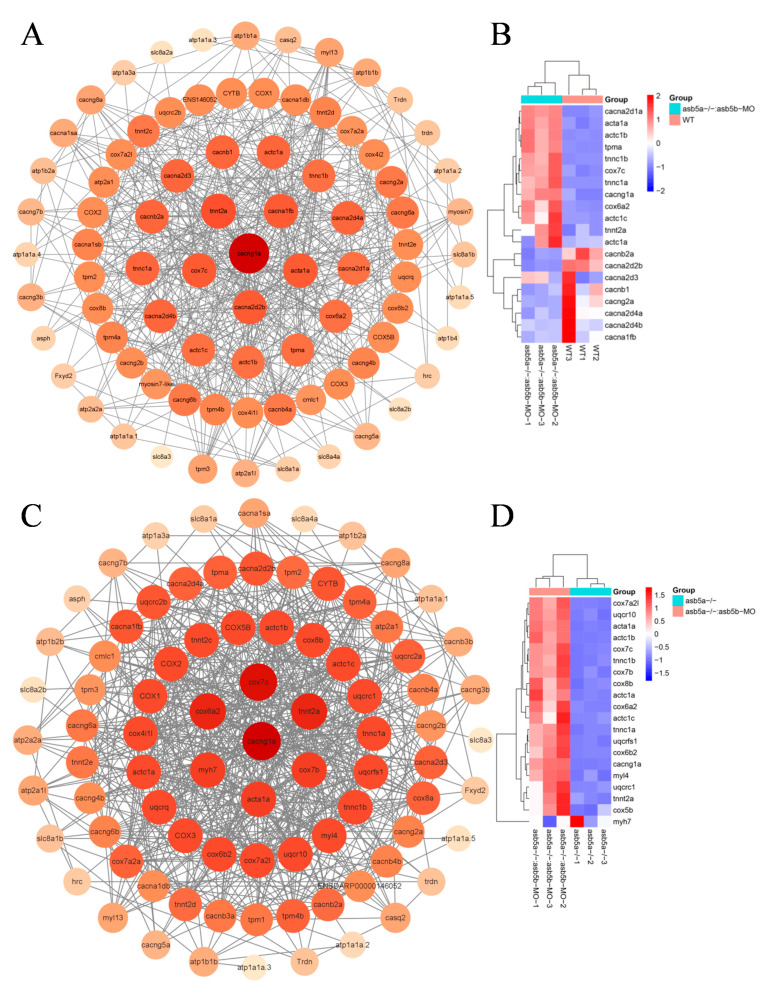
PPI network analysis and heatmap of the top 20 hub genes. (**A**) PPI regulatory network between the *asb5a^−/−^:asb5b*-ATGmo double-knockout and WT groups. All genes were enriched in cardiac contraction entries in KEGG, including 85 nodes and 499 edges. Larger circles and darker colors indicate higher degrees of connectivity. (**B**) Heatmap of the top 20 hub genes between the *asb5a^−/−^:asb5b*-ATGmo double-knockout and WT groups. (**C**) PPI regulatory network between the *asb5a^−/−^:asb5b*-ATGmo double-knockout and *asb5a^−/−^* groups. All genes enriched were in cardiac contraction entries in KEGG, including 86 nodes and 576 edges. Larger circles and darker colors indicate higher degrees of connectivity. (**D**) Heatmap of the top 20 hub genes between the *asb5a^−/−^:asb5b*-ATGmo double-knockout and *asb5a^−/−^* groups. Rows and columns represent genes and samples, respectively. Red and blue represent high and low expression levels, respectively. Darker colors indicate more pronounced differences.

**Figure 11 ijms-24-16364-f011:**
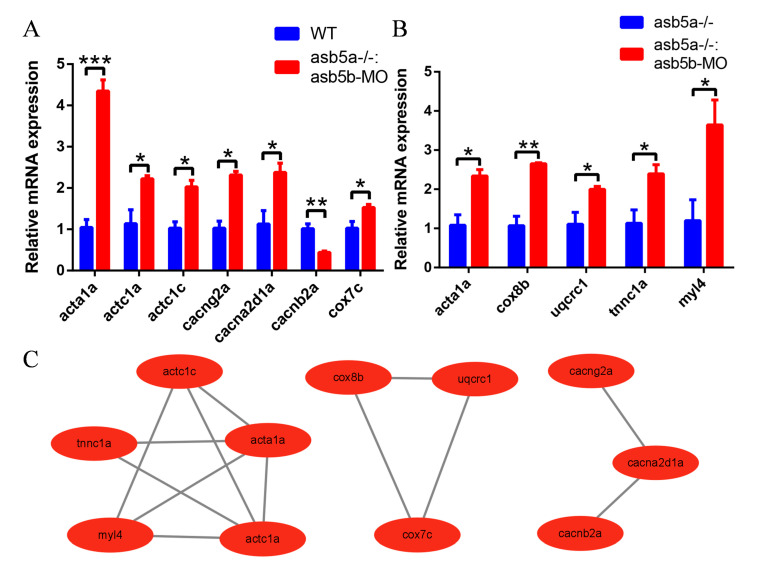
qPCR validation and the PPI network diagram. (**A**,**B**) Relative qPCR quantification of hub genes’ mRNA levels in zebrafish hearts at 48 hpf. Error bars indicate SEM. * *p* < 0.05; ** *p* < 0.01; *** *p* < 0.001 (N = 3 and n = 10; unpaired *t*-test). All genes were tested with 3 sample replicates (N = 3), each sample containing 10 embryos (n = 10). (**C**) PPI network containing three modules with 11 nodes and 13 edges.

**Figure 12 ijms-24-16364-f012:**
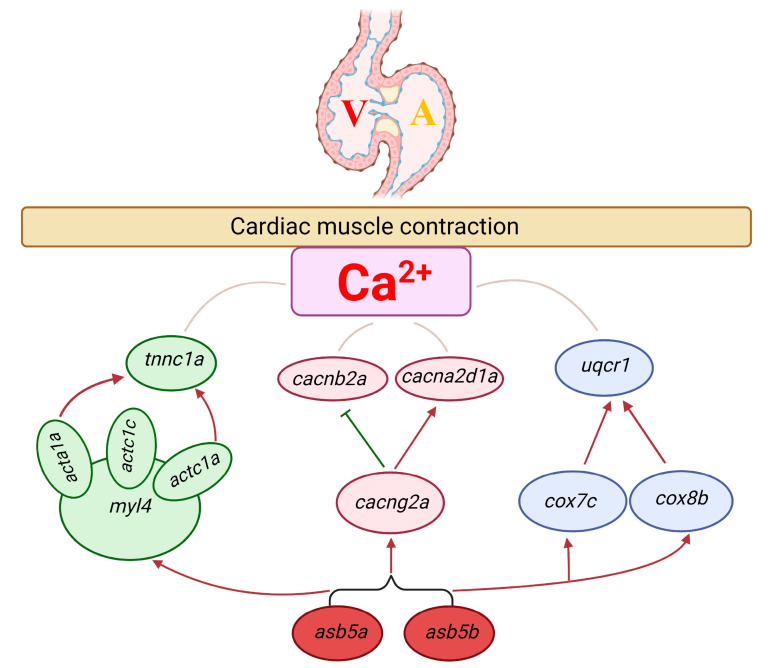
Probable molecular regulatory mechanism of 11 hub genes affecting cardiac contractile function following*asb5a^−/−^:asb5b*-ATGmo double-knockout. Green, pink, and blue ovals represent the three different control modules, respectively. Red and green arrows represent positive and negative regulation, respectively.

## Data Availability

All data generated or analyzed during this study are included in this published article and its supplementary information files.

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
