# Peer review of "asb5a/asb5b Double Knockout Affects Zebrafish Cardiac Contractile Function"

_ijms, 2023, doi:10.3390/ijms242216364_

Round 1

Reviewer 1 Report

Comments and Suggestions for Authors

Cai et al study the concurrent knockout of the asb5a and asb5b genes in zebrafish. They used transcriptomics to show that the kos impact early cardiac contractile function. They use semi-automatic heartbeat analysis to confirm their results in vivo. They enrich their study by using PPI Network Construction and Hub Gene Screening and validate 20 Abnormally Regulated Hub Genes using qPCR.

in 4.5. RNA-seq–Based Transcriptome Analysis, more details should be given concerning normalization

in 4.8 the name of the normalizing gene used for qPCR should be indicated and its use justified.

Author Response

  1. in 4.5. RNA-seq–Based Transcriptome Analysis, more details should be given concerning normalization

Answer 1:

Thank you for your kind advice. In 4.5. We have added more standardized details in RNA-seq-based transcriptome analysis.

  1. in 4.8 the name of the normalizing gene used for qPCR should be indicated and its use justified.

Answer 2:

Thank you for your kind advice. In 4.8, the name of the normalizing gene used for qPCR is actb1. Target gene expression was normalized using the internal control actb1 (also named β-actin, an internal reference gene, which has a relatively constant expression in various tissues and cells, and is often used as a reference when detecting gene expression level changes). (This section has been added to “Materials and Methods 4.8”).

Reviewer 2 Report

Comments and Suggestions for Authors

The authors in the present study have evaluated the role of a subset of ubiquitin ligase E3- asb5 in cardiac development and function of zebra fish. In zebrafish, the asb5 have two subtypes: asb5a and asb5b. The authors have found that simultaneous asb5b knockdown in asb5a-knockout groups led to severe pericardial cavity enlargement and atrial dilation. To further explore the mechanism, they did transcriptome analysis and came up with top 11 hub genes having potential role in cardiac development of zebrafish under regulation of asb5a and asb5b. Their finding is interesting and holds significance for publication in IJMS. My concerns are following.  

1.     What is the homologous gene for asb5 in mouse and human. Have the cardiac development and function been assessed before with these two subtypes (asb5a and asb5b) in mouse or human. If not, please write the explanation in introduction. Please make the rational or objective behind the study clear as it is not clear how asb5 an ubiquitin ligase impacting cardiac development in zebra fish. Please write the rational/objective behind this study more specifically in the introduction.

2.     Please draw a schematic diagram or label in zebrafish from blastocyst stage to 4 dpf reflecting the notochord, eyes and head regions for the readers who don’t have zebra fish research background and are still interested in the function of asb5 gene/ protein in cardiac development and function.

3.     Figure 4D’ and 4E’, as compared to WT (Figure 4A’) there is a significant enlargement of pericardial cavity in asb5b KO group alone at 48hpf when 0.75 and 1mM dose of morpholino oligos was used. How will you explain this?

4.     Please explain Figure 4A’’–4G’’ as it is not clear how myl7: EGFP is indicating cardiac development in zebrafish

5.     Line 242-245: Please explain what is Fractional shortening (FS) and how it impacts cardiac functions. Is there any data of cardiac output?

Author Response

  1. What is the homologous gene for asb5 in mouse and human. Have the cardiac development and function been assessed before with these two subtypes (asb5a and asb5b) in mouse or human? If not, please write the explanation in introduction. Please make the rational or objective behind the study clear as it is not clear how asb5 an ubiquitin ligase impacting cardiac development in zebrafish. Please write the rational/objective behind this study more specifically in the introduction.

Answer 1:

By consulting the database, we found that the asb5 gene is highly homologous in higher vertebrates such as human, mouse and rabbit, and it is a single copy gene and is not typed[2]. In zebrafish, however, the asb5 gene is divided into two subtypes, asb5a and asb5b. There are no functional study reports of asb5 gene on heart development in mouse or human so far;( This section has been added to the Introduction).  It was reported that asb11 knockdown in zebrafish caused an obvious enlarged pericardial cavity and Asb gene family transcriptome analysis indicated asb5 associated with the development of skeletal muscle [reference 3 and 4]. All of these preliminary studies provide some clues that ASB5 may have a role in heart development. ( This section has been added to the Introduction). Based on the above two points, we carried out this work.

  1. Please draw a schematic diagram or label in zebrafish from blastocyst stage to 4 dpf reflecting the notochord, eyes and head regions for the readers who don’t have zebrafish research background and are still interested in the function of asb5 gene/ protein in cardiac development and function.

Answer 2:

Thanks for your kind reminder. We have labeled the notochord, eyes, and head regions with red triangular arrows and capital letters in Figure2 and Figure3.

  1. Figure 4D’ and 4E’, as compared to WT (Figure 4A’) there is a significant enlargement of pericardial cavity in asb5b KO group alone at 48hpf when 0.75 and 1mM dose of morpholino oligos was used. How will you explain this?

Answer 3:

Thank you for your carefulness. Indeed, as Figure 4D’ and 4E’ showed, there were a trend in atrial dilatation in asb5b KO group, yet the difference did not reach statistically significance as Figure 9C showed.

  1. Please explain Figure 4A’’–4G’’ as it is not clear how myl7: EGFP is indicating cardiac development in zebrafish.

Answer 4:

    Thanks for your kind reminder. myl7 is specially expressed in the heart and myl7: EGFP is used as a cardiac specific expression marker. ( This section has been added to the Result 2.3 and Methods 4.1)

  1. Line 242-245: Please explain what is Fractional shortening (FS) and how it impacts cardiac functions. Is there any data of cardiac output?

Answer 5:

Fractional shortening (FS) represents the ejection volume at each contraction of the heart. The reduced FS indicates the inability of the heart to contract completely during systole, corresponding to systolic heart failure in human adult heart(This section has been added to the result 2.5). The Excel data for detailed output will be sent as an attachment to the editor, who will then send it to you (as only Word or PDF can be uploaded here).